# Perceived barriers in accessing sexual and reproductive health services for youth in Lao People's Democratic Republic

**Souksamone Thongmixay** [1]*, **Dirk Rombout Essink**[2], **Tim de Greeuw**[2], **Viengnakhone Vongxay**[1], **Vanphanom Sychareun**[1], **Jacqueline E. W. Broerse**[2]

**1** University of Health Sciences, Vientiane, Lao PDR, **2** Vrije Universiteit Amsterdam, De Boelelaan, the Netherlands

* souksamone_yod@yahoo.com

## Abstract

### Background

Sexual activity during youth is common in Lao PDR. However, young people seldom utilize sexual and reproductive health services and subsequently suffer from poor sexual and reproductive health. The aim of this qualitative study was to explore the barriers perceived by youth that prevent their access to sexual and reproductive health services.

### Methods

Twenty-nine semi-structured interviews were conducted with 22 participants aged 15–25 years, from urban and rural areas. A vignette was used during interviews with those who had no experience with sexual and reproductive health services. Additionally, seven semi-structured interviews were conducted with health providers from youth-friendly health clinics and from public sexual and reproductive health services. Data were analyzed using a thematic approach.

### Results

The main barriers preventing young people from accessing sexual and reproductive health services were related to *cognitive accessibility* and *psychosocial accessibility*. The *cognitive accessibility* barriers were a lack of sexual knowledge and a lack of awareness of services. Perceived barriers in *psychosocial accessibility* were the feelings of shyness and shame caused by negative cultural attitudes to premarital sex, and the fear of parents finding out about visits to public sexual and reproductive health services, due to lack of confidentiality in the services and among health providers. In addition, the barriers of *geographical accessibility*, mainly insufficient availability of youth-friendly health clinics.

### Conclusion

To improve access to services, a multi-component strategy is needed: promotion of youth-friendly health clinics; sexual education in schools; a formal referral system between schools

---

**Data Availability Statement:** Data are within the article and its supporting information files. Verbatim quotes from participants cannot be

shared because they are potentially identifying. This data are available upon request to the following contact: Faculty of Public Health, University of Health Sciences, P.O. Box 7444, Samsanthai Road, Vientiane, Lao PDR. Tel and Fax +8562124085.

**Funding:** The research was conducted under the supports of European funded LEARN Project (Number: DCI/SANTI/2014/342-306) and the Medical Committee of Netherland-Vietnam (MCNV) in Lao PDR. The first author is the principal.

**Competing interests:** The authors have declared that no competing interests exist.

and youth-friendly health clinics; and community support interventions. Prior to implementation, more research should be done on the applicability of these methods in the Laos context. Future research should try to determine the cost-effectiveness of youth-friendly health clinics integrated in a district hospital and stand-alone clinics, to provide insight into which form should be further developed.

## Introduction

Among the 1.15 billion adolescents in the world, more than 700 million live in Asia [1]. Youth (15–24 years) comprise about third of the population in Asia. Globally, SRH problems are one of the main causes of high morbidity and mortality rates among young people [2]. Aside from the contextual causes, the disproportionate burden of SRH issues that young people suffer is often aggravated by the lack of SRH knowledge and low availability and/or accessibility of SRH services [3]. The accessibility of SRH services is influenced by a complex set of factors related to: youth's SRH knowledge and awareness of services, socio-cultural norms regarding sexual activity of youth, availability of services, costs of using the services and the quality the services they provide [4].

In the Lao People's Democratic Republic (Lao PDR), sexual and reproductive health (SRH) care can be weak; especially young people suffer from poor SRH. Lao PDR has the highest adolescent fertility rate in Southeast Asia, with 83 births per 1000 women aged 15 to 19 years [5]. Furthermore, early sexual debut is common [6], at an age when SRH knowledge is lacking, SRH literacy is poor, [7,8] and risky sexual behaviours, such as low condom use and having multiple partners [5] occur, leading to a high risk of STIs including HIV [9]. The public SRH in Lao PDR are only used by married adult women who already have children [10]; unmarried youth without children do not access these services, further increasing their risk for unplanned pregnancies and untreated STIs. In 2011, the Lao Women's Union, with funding from UNFPA, set up the 'Vientiane Health Centre for Youth and Development', a stand-alone clinic providing gender specific counselling, STI treatment, and modern contraceptives for a small fee. When the project funding ended in 2015, they continued to offer services at youth-friendly health clinics (YFHC) in Vientiane and Savannakhet, supported by income from fees.

Several factors may increase adolescents' risk of adverse SRH events and preventing access to appropriate SRH services [11, 12]. Multiple studies have focused on barriers that hinder youth from accessing SRH services [4, 13, 14]. Currently, there are only two Youth Friendly Health Clinic (YFHCs) in Laos. In both, staff were trained in youth-friendly service delivery and related service protocols, and in recruitment and training of medically-trained volunteers to provide SRH information and services. In Lao PDR, there is no data on the perceptions of the most relevant stakeholders in this matter: the health providers and the youth themselves. As a theoretical model, we adapted the definitions and conceptual frameworks of Bertrand, Hardee, Magnani, & Angle [15], Levesque et al [16] and the United Nations Committee on Economics, Social, and Cultural Rights (UNCESR) [17]. The final conceptual model included five key concepts: *cognitive accessibility*—or knowledge and awareness; *psychosocial accessibility*—reflecting cultural and social norms affecting youth's use of SRH services; *geographic accessibility*–whether the facilities are within the area youth can travel to; *affordability*–whether youth can pay for the or not; and finally, q*uality of services*- whether the services are appropriate for youth as clients [18].

This study aimed to explore perceptions of barriers to accessibility of SRH services for youth, to stimulate evidence-informed decision-making in health policy formulation and implementation. In this way, we hope to contribute to making Lao SRH services more accessible to youth.

## Methods

### Study design

Four participant groups were identified: (i) youth who had never used SRH services (ii) youth having experience with SRH services (iii) health providers from public SRH services and (iv) health providers from Youth Friendly Health Clinic (YFHCs). Qualitative research methods were used to investigate their perceptions regarding accessibility of SRH services for youth and on solutions to improve the current situation.

### Study site

Data were collected in two provinces of Lao PDR, which were purposively selected based on availability of YFHCs: Vientiane (VTE) and Savannakhet (SVK). VTE is the capital city, with 91,575 adolescents aged 15–19 years and 26 births per 1000 girls. SVK, out of a total adolescent population age 15–19 years of 93,351, recorded 69 births per 1000 girls [5]. To investigate the effect of living area on the barriers preventing youth access to SRH services, in each province one urban and one rural district were selected. The urban districts Sikhottabong and Kaysone were chosen based on proximity to the YFHCs. The rural districts Hatsayfong and Songkhone were chosen based on proximity to the Thai border; previous research showed that many Laotians living near the border crossed it to use Thai health care services because they considered the quality of Thai health services better than in Laos [19].

### Study population

Youth were eligible to participate in the current study if they (a) were between 15–25 of age; (b) either had experience or had no experience with SRH services or YFHC; (c) lived in VTE and SVK; (d) had resided in the VTE and SVK at least 12 months; and e) provided signed informed consent. Health providers were eligible to participate if they (a) were working in public SRH services and YFHCs as their regular work for at least 1 year; (b) provided signed informed consent.

### Sample size and sampling technique

Youth and health providers participating in this study were recruited in diverse ways. First, the research team worked with village health volunteers (VHV) to find youth who had no experience with SRH services. The VHV asked youth in their community to participate in the study; if they were interested, the researchers came to conduct the interviews. Second, purposive sampling [20] was used to recruit adolescents who did have experience with YFHCs. In Vientiane, the researchers sat in the clinic waiting room for two days and invited youth visiting the YFHC site to participate. If they were interested, interviews were conducted in a separate consultation room of the facility. Researchers emphasized that refusal to participate would have no influence on the care they received. In the YFHC in Savannakhet, the health provider informed youth attending the clinic about the research. Appointments were made with those interested to conduct the interview in the following days in the YFHC. Youth between 15 and 25 years of age were recruited. Third, health providers from both public SRH services and YFHCs whose daily work involved contact with care recipients were invited through the university network,

with the approval of the health facility head. All participants had to speak the Lao language. Sample size reflected saturation reached based on the research questions. A previous qualitative study recruited 30 adolescents, married and unmarried [21]. We expected a similar number for the current study. However, we found that the information reached saturation at 22 participants interviewed, no new answers/information arose. Only four non-experienced adolescents were recruited; they all provided similar information so we did not try to recruit more. There were 7 health care providers because we selected one from each facility, based on experiences from previous studies.

## Data collection tools

A semi-structured interview guide was developed for each participant group. The interview guides for all youth were divided into five sections, based on the concepts of the theoretical model. The definitions and conceptual frameworks of Bertrand, Hardee, Magnani, & Angle [15], Levesque et al [16] and the United Nations Committee on Economics, Social, and Cultural Rights (UNCESR) [17] were adapted to produce a merged conceptual model consisting of five key concepts: *cognitive accessibility*—the extent to which youth are aware of the existence of SRH services and their perceived SRH needs, including knowledge regarding contraceptive methods and risks for pregnancies and STIs when having sexual intercourse; *psychosocial accessibility*—the extent to which cultural and social norms influence youth to use SRH services and their perceived agency and autonomy to make decisions by themselves; *geographic accessibility*—the extent to which SRH services can be reached physically and the amount of time it takes; *affordability*—the perceived agency of youth to pay for reaching and using SRH services; and finally, *quality of services*—the extent to which the provided SRH services match with the perceived needs of youth: accessible, acceptable and appropriate [18]. Every section included open-ended questions, such as "*What do your parents think of premarital sex*?" Every section ended with an open question about how the barriers mentioned in the discussion could be overcome. Because sexuality is a sensitive topic in the Lao culture, two structured vignettes [22], one for each gender's perspective, were developed for the interviews with youth who had never used SRH services. The vignettes are short fictional stories about two sexually active adolescents who want to obtain SRH services. Participants were asked to give their opinion or advice about what the fictional adolescents should do. The interview guides for health providers of both public SRH services and YFHCs consisted of open-ended questions focusing broadly on their opinions on SRH needs of Lao youth, barriers preventing youth to access SRH services, and their solutions to remove these barriers.

## Data collection procedures

Finally 29 semi-structured, in-depth interviews were conducted, involving 22 young people and seven health providers at both public SRH services and YFHCs. All but four participants were interviewed individually. For logistic reasons, four young people from Kaysone district in Savannakhet, who had no experience with SRH services, were interviewed in sets of two, each set of the same gender. Vignettes were used during the first half of the interview with youth who had no SRH experience, after which sufficient rapport had been built with the participant that the vignettes were no longer needed. Prior to interviews, the purpose and procedure were explained, and a confidentiality agreement was explained and verbally agreed to. All 29 participants agreed to the interviews being recorded. Teams of one Dutch and one Lao researcher conducted all interviews, with the Lao researcher translating as necessary. Two native Lao speakers trained in qualitative research methods (the first and fourth authors) conducted all interviews in Lao using the same semi-structured interview guide, which included open-ended

questions and probes. As sexuality is a sensitive topic in Lao culture, participants and interviewers were matched for gender. Interviews took place in locations with as much privacy as possible, such as closed rooms or quiet places in the community. The interview duration ranged from 45 to 100 minutes. Both youth and health providers received small gifts after the interviews as an appreciation for their participation.

## Ethical considerations

Ethical approval for the study was granted by the Ethics Committee for Health Research of the University of Health Sciences, Lao PDR. Among the participants, half were under the age of majority in Lao PDR, that is, under 18. This study was part of the project "Unmet need for family planning: New approach model, Lao PDR". Parental consent was not required by the Committee because in Lao PDR minors, considered to be adolescents 15 to 18, can give consent themselves. Parental consent would be needed for children under 15 years of age. The three youngest interviewees were 15, others were 16 or 17 and all confirmed their consent to participate the study. The study carried minimal risks for participants and the interviews were done in safe and private locations.

Before each interview, we gave every interviewee a one-page information sheet, which described the purpose of study. We introduced ourselves and explained about consent and confidentiality, that participation was voluntary and anonymous and that they had the right to refuse to answer any question or end the interview without explanations and without any consequences to them. To guarantee confidentiality, names were replaced with a code in all recordings and any identifying information was removed from all records.

## Data analysis

The recordings were transcribed verbatim, summarised and thematically analysed using a deductive approach [23]. Transcripts were translated into English, then read and re-read by two researchers to identify and discuss emerging themes among the research team, to increase validity. Thereafter, all transcripts were coded. Thematic coding [23] was based on the key concepts of access to healthcare; similar codes were categorised and clustered in sub-themes. These formed the initial coding frame, broadly related to (i) current SRH and health seeking behaviour (ii) barriers to accessing SRH services, and (iii) solutions to improve access to SRH services. Within these sub-themes, findings were compared according to participant characteristics (gender, living area, SRH service use/non-use) by the researchers and experts in SRH.

## Results

### Characteristics of participants

Finally 22 young people and seven health providers (one from each YFHC, and five from public SRH services) were included in the study. Among the young people, 14 had no experience with SRH services, two had experience with general SRH services from a public hospital, and six were recruited through the YFHC, with which they had experience. The mean age of youth participants was 18.5 years (range 15 to 25). Approximately one-third of the youth was in secondary school at the time; of the eight who had completed secondary school, half continued their studies at university (Table 1). One female participant had an unplanned pregnancy at the age of 14 and dropped out of secondary school. One male adolescent was homosexual, all other youth perceived themselves as heterosexual. Half of the 22 young people reported being sexually active, almost all of whom were married or in a union (Table 2).

**Table 1. Socio-demographic characteristic of the interviewed key informants.**

| Types of key information | Adolescent girl | Adolescent boy | Health provider |
|---|---|---|---|
| **Age** | | | |
| 15–19 | 8 | 7 | 0 |
| 20–25 | 2 | 5 | 0 |
| 25–49 | 0 | 0 | 5 |
| > = 50 | 0 | 0 | 2 |
| **Sex** | | | |
| Male | | 12 | 1 |
| Female | 10 | | 6 |
| **Education** | | | |
| Primary | 0 | 0 | 0 |
| Lower SS | 5 | 4 | 0 |
| Upper SS | 5 | 5 | 0 |
| College/University | 0 | 3 | 7 |
| **Had sexual encounter** | | | |
| Yes | 6 | 6 | - |
| No | 4 | 6 | - |
| **Experience with SRH services** | | | |
| Yes | 4 | 4 | - |
| No | 6 | 8 | - |
| **Providers' Work place** | | | |
| YFHC | - | - | 2 |
| Public SRH services | - | - | 5 |

## Current sexual and reproductive health attitude and behaviour

### Attitude.

Most of the youth were positive about sexual activity at their age, found it acceptable. However, three male and four female participants indicated that they wanted to postpone sexual activity at least until they finished schooling. Not feeling ready, fear of health risks, consequences for school and future life, and the fear of disappointing their parents were frequently mentioned reasons to postpone sexual activity.

### Sexual behaviour.

Half of the young people reported being sexually active. Four male participants indicated that they had more than one sexual partner, but said their partners were not aware of that fact. Subsequently, several female participants noted their uncertainty about the loyalty of their partner.

**Contraceptive use.** Young people who were sexually active all wanted to prevent pregnancies and used both modern contraceptives (MC) and traditional contraceptive methods. The most common MC were the condom, the contraceptive pill and the emergency pill. However, the majority admitted not using MC every time they had sexual intercourse, and regularly using traditional contraceptive methods, such as withdrawal or periodical abstinence, instead.

Reasons given for not using MC were: not having MC readily available, forgetting to use MC because of alcohol consumption, preferring a more natural sensation without MC,

**Table 2. Brief descriptions of youth participants.**

| | |
|---|---|
| **Respondent 1** | Eighteen-year-old male living in urban Vientiane. He was in seventh grade at private secondary school, had a girlfriend, was sexually active, and had not used any SRH services before. |
| **Respondent 2** | Eighteen-year-old male living in urban Vientiane. He was in seventh grade at private secondary school, was single and had no sexual experience. |
| **Respondent 3** | Twenty-three-year-old male living in urban Vientiane. He studied at university and had a part-time job as a construction worker. He had a girlfriend and was sexually active. He used the YFHC for STI testing and treatment. |
| **Respondent 4** | Twenty-year-old male living in urban Vientiane. He studied at university and had a part-time job as a waiter in a restaurant. He had a girlfriend and was sexually active. He used the YFHC for STI testing and treatment. |
| **Respondent 5** | Twenty-three-year-old male living in urban Vientiane. He had finished secondary school and worked at a phone store. He had a girlfriend, was sexually active and used the YFHC for STI testing and treatment. |
| **Respondent 6** | Sixteen-year-old male living in a rural village in Vientiane. He was in sixth grade of secondary school. He was single and had no sexual experience. |
| **Respondent 7** | Seventeen-year-old male living in a rural village in Vientiane. He was in sixth grade of secondary school, was single and had no sexual experience. |
| **Respondent 8** | Fifteen-year-old male living in urban Savannakhet. He was in first grade of secondary school, was single and had no sexual experience. |
| **Respondent 9** | Seventeen-year-old male living in urban Savannakhet. He was in third grade of secondary school, was single and had no sexual experience. |
| **Respondent 10** | Twenty-five-year-old male living in urban Savannakhet. He had just finished his studies at university and was job-seeking. He was a homosexual, had a boyfriend and was sexually active. He used the YFHC for STI testing. |
| **Respondent 11** | Twenty-year-old male living in a rural village in Savannakhet. He finished secondary school and was job-seeking. He had a girlfriend, was sexually active and had used condoms from drugstores. |
| **Respondent 12** | Fifteen-year-old male living in a rural village near the district hospital in Savannakhet. He was in first grade of secondary school, was single and had no sexual experience. |
| **Respondent 13** | Eighteen-year-old female living in urban Vientiane. She was in seventh grade of a private secondary school, was single, and had no sexual experience. |
| **Respondent 14** | Seventeen-year-old female living in urban Vientiane. She was in seventh grade of a private secondary school. She was single and had sexual experience but had not used any SRH services. |
| **Respondent 15** | Sixteen-year-old female living in a rural village in Vientiane. She was in fourth grade of secondary school, was single and had no sexual experience. |
| **Respondent 16** | Seventeen-year-old female living in a rural village in Vientiane. She was in sixth grade of secondary school. She had a boyfriend and was sexually active but had not used any SRH services. |
| **Respondent 17** | Fifteen-year-old female living in urban Savannakhet. She was in first grade of secondary school, was single and had no sexual experience. |
| **Respondent 18** | Eighteen-year-old female living in urban Savannakhet. She was in sixth grade of secondary school, was single and had no sexual experience. |
| **Respondent 19** | Twenty-four-year-old female living in urban Savannakhet. She had finished secondary school and had a job. She got married last year and reported being sexually active since the age of 16. She used the YFHC to obtain an injectable contraceptive. |
| **Respondent 20** | Twenty-two-year-old female living in urban Savannakhet. She had finished secondary school and worked in a factory. She had a boyfriend and was sexually active. She used the YFHC for STI testing and treatment. Her boyfriend did not want to be tested as he had no symptoms of an STI. |
| **Respondent 21** | Sixteen-year-old female living in a rural village in Savannakhet. At the age of 15 she had an unplanned pregnancy and dropped out of secondary school. After delivery she moved in with her parents-in-law. She had used SRH services from the district hospital after delivery. |
| **Respondent 22** | Seventeen-year-old female living in a rural village in Savannakhet. She had finished secondary school and worked as housekeeper. She had a boyfriend, was sexually active and used SRH services in a district hospital but in a different district from her place of residence. |

perceiving traditional contraception as a safe alternative for MC, and refusal of partner to use MC. One male participant, who had multiple sexual partners, told us that he did not use MC with his partner to show her respect.

Responses about who made the decision on condom use were inconsistent. Although both male and female participants thought it should be a joint responsibility, the majority of sexually active female participants indicated that it was always their partner who made the decision whether to use condoms. Although these female participants preferred to use a condom, this did not always happen. They perceived this as a matter of course and did not dare to oppose their partners.

**Use of SRH services.**   SRH services were provided in public SRH units (located in district and provincial hospitals), YFHCs, and private units (private clinics, pharmacies). Most young people could identify at least one of the available SRH services in their community, with pharmacies mentioned and utilised most often. Youth who knew the YFHCs had been informed either by a volunteer or through their peer network. Health providers in YFHCs encouraged visitors to tell friends about the clinic.

### Information seeking.

Adolescents obtained SRH information through various channels. Besides Internet and social media, most information was acquired through peer education. Leaflets in contraceptive packages were mentioned as a reliable source of information, but in practice they were hardly used.

## Barriers to access the sexual and reproductive health services

The barriers adolescents perceived with regard to accessing SRH services are described in the order of the theoretical framework: cognitive accessibility, psychosocial accessibility, geographical accessibility, affordability and the quality of the services.

### Cognitive accessibility.

There was a discrepancy between youth's stated beliefs and their behaviour in using SRH services. Although they reported having insufficient knowledge about SRH, they did not seek information from SRH services. When specifically asked, most young people seemed to know some form of MC and were aware of the risks of pregnancy and STIs. However, their knowledge was insufficient, sometimes leading to health risks and unwanted pregnancies.

About half of the participants also had inadequate risk perception regarding unsafe sex. They only sought help from SRH services when they experienced a problem with their SRH; they did not try to obtain information before a problem arose.

All but one of the adolescents had had at least a rudimentary form of SRH education in school or during extracurricular activities. Nevertheless, lack of appropriate education was mentioned as a reason for their insufficient knowledge about MC and the risk of pregnancy and STIs. In secondary schools, SRH education was integrated in biology courses. Adolescents complained that the course focused on body functions, with insufficient emphasis on knowledge and practical skills related to their SRH. The majority of adolescents suggested a need for more education about MC and available SRH services, and about how to talk about SRH issues. The respondents living in rural areas reported more often that SRH education lacked information on contraception or did not cover SRH services. Health providers also mentioned that adolescents living in rural areas had poorer SRH knowledge than their urban peers.

The two youth who had attended general SRH services said that they only got information when they specifically asked for it; health care providers did not offer them any sexual health information.

In addition to the inadequacies of SRH education, adolescents indicated it was difficult to find SRH information in Lao language on the Internet, so they sought information in Thai. Also, some found it difficult to identify reliable websites.

Another barrier related to the cognitive accessibility was lack of awareness of where to obtain SRH services. The youth respondents believed that public FP units could only be used by married individuals and not by single adolescents.

The YFHCs were not widely known, even among adolescents living in the same district as the clinic; those who did not use the YFHC had never heard of its existence. According to a physician working at the YFHC in Vientiane, this was mainly due to the poor promotion of the clinic.

In both Vientiane and Savannakhet, youth-friendly clinics were poorly visible from the public road. In Vientiane, only an inconspicuous sign identified the facility. In Savannakhet, the clinic was only indicated on the property of the district hospital.

**Psychosocial accessibility.**

Another barrier was related to psychosocial accessibility. A perceived negative cultural attitude towards sexual activity before marriage was especially felt by females. Parents actively discouraged relationships among youth, and topics like relationships and sex were taboo to discuss with parents.

Despite the cultural barriers to access to SRH services, both health providers and adolescents mentioned that a cultural transition was underway: the current adolescent generation was more open and accepting towards premarital sex, compared to older generations.

Shyness and shame caused by the cultural lack of acceptance of premarital sex were mentioned as the main reasons that adolescents felt reluctant to access SRH services and to talk about SRH with a health provider. This affected females more than males. However, they were mainly afraid of being seen by others from the community when accessing the SRH services. Aside from negative comments from community members, adolescents feared that if someone saw them, their parents would be informed, would be disappointed in them and may get angry.

Public FP units were seldom used because of this fear and the lack of privacy in these facilities. The services were located at the Department of Mother and Child Health (MCH), only used by pregnant women and women with babies felt. Adolescents felt very noticeable sitting in the waiting area and thought everyone would know why they were there. Health providers acknowledged the lack of privacy in the waiting areas.

The majority of adolescents did not report fearing negative attitudes of health providers in public SRH services. Even though health providers are from an older generation, adolescents assumed that they were professionals who would not treat adolescents in a negative way. Nevertheless, adolescents thought that health providers working at YFHCs were friendlier, understood their problems better and would make more time for them.

Even though adolescents did not fear negative attitudes from health providers, they did doubt their confidentiality, and feared that health providers might inform their parents if they accessed SRH services. This was especially true in rural areas, where health providers and the community know each other well. To overcome the lack of privacy and confidentiality, four adolescents had visited SRH services outside their own district.

Most health providers were not to be in favour of premarital sex, but thought that cultural change was inevitable. Moreover, they considered the adolescents' health more important than

cultural norms and would never be negative when treating health-seeking adolescents. Health providers were, however, worried that their work would be perceived by the community as promotion of premarital sex and were therefore sometimes reluctant to provide SRH information to adolescents. Health providers were familiar with the rumours about the lack of confidentiality, but denied they were true.

**Geographical accessibility.**

Geographical accessibility was not perceived as a barrier to access SRH services. However there was mention of a geographical barrier which was the distance of travel to the service site. Adolescents mentioned that they would go across the border to Thailand for service rather than going to the youth clinic which was too far for them. Adolescents living in the rural district Songkhone were an exception, indicating that their district hospital was hard to reach due to poor infrastructure. Therefore, they preferred to obtain their SRH services in Thailand.

Although limited number of service sites was not mentioned as a major barrier, both adolescents and health providers indicated there were insufficient health facilities specifically targeting adolescents: the two existing YFHCs are clearly insufficient for the whole country. Furthermore, the existing YFHCs did not fully match the needs of healthcare seeking adolescents. Adolescents preferred to obtain services in evenings or weekends but the YFHC in Savannakhet was only open during office hours. The Vientiane service was open seven days a week from 9 AM till 5 PM. Health providers of both services solved this by being available by phone in their free time.

**Affordability.** Costs related to reaching and using the SRH services were also not noted as a major barrier. However, youth who had no experience with SRH services did not know that the services were practically free. Most estimated the service fees to be significantly higher than the actual 5000 LAK (€0.50).

## Quality of services

Youth-friendly services aim to provide quality of care that is accessible, acceptable, and appropriate for adolescents. Privacy and confidentiality were mentioned by many participants as being important components of YFS. Youth did not report issues with the quality of provided services (apart from the earlier mentioned confidentiality and privacy issues) as a barrier to access either YFHCs or public SRH services.

The different SRH services were accessed for different purposes. Firstly, public FP units were mainly used for pregnancy care and contraception services for married couples. Youth perceived these services as the least comfortable for them to use for other SRH issues, mainly because of lack of privacy and to their perception that the services are for married women with children. Health providers in these facilities reported that they rarely have adolescents as clients.

Secondly, youth who attended YFHCs mostly came for STI counselling and testing when they had symptoms of an STI. Only a few came for counselling or for contraceptives. At the time of data collection, the YFHC in Vientiane was seldom visited by female clients. According to the male physician there, the reason was the temporary absence of a female physician who was on study leave. The YFHC in Savannakhet was used by both male and female clients, albeit rarely by females.

Lastly, pharmacies were a source of contraceptives and emergency/abortion pills, with limited or no counselling. Adolescents also mentioned going to pharmacies to get drugs for self-treatment of STIs. Adolescents experienced access to pharmacies as easiest, because of the perceived anonymity, the favourable opening hours, and the number of services available, especially in urban areas.

## Discussion

The results of this study provide an insight into adolescent sexual behaviour and barriers related to *cognitive accessibility* and *psychosocial accessibility* of SRH services in Lao PDR. The narratives of our respondents clearly showed that a proportion of the youth is sexually active but are often not taking appropriate precautions to promote their and their partners' SRH. The preventive measures that youth in this study used–often irregularly—were condoms, oral pills and emergency pills, as well as traditional contraceptive methods, such as withdrawal or periodic abstinence. Most unmarried youth said rather than using public health facilities, they looked for SRH services from private clinics and pharmacies.

The respondents appeared poorly informed about reproductive health. Most did not recognize the health and social risks associated with unprotected sex, early marriage, early pregnancy and childbirth, and STIs. These findings correspond with those of a review [24] pointing out that in Vietnam and other parts of Asia, adolescents and youth face reproductive health challenges, including early pregnancy, which is mostly unwanted, and complications of unsafe abortion, but have limited access to good quality and friendly SRH care, covering STI and safe abortion services.

Our results also show that adolescent girls who have sex do not always have the power of decision about the use of MC. Such power differences have been described before in Laos, in a study on determinants of adolescent pregnancy and access to SRH services, which recommended that efforts should be increased to provide knowledge and communication skills to adolescent girls so that they can take more control in contraceptive decision-making [21].

### Cognitive accessibility

A first barrier related to cognitive accessibility was knowledge of SRH. Earlier studies in Lao PDR also reported that youth lacked SRH literacy [8] and that they perceived themselves to be at low risk for STIs and unwanted pregnancies [25, 7]. Vongxay et al indicated that more than 65% of in-school adolescents in Lao PDR had inadequate SRHR health literacy [8]. Lack of knowledge and health literacy has been demonstrated to lead to less demand for SRH services [26, 27]. The importance of improving young people's SRH knowledge was also appreciated by the youth, who called for more SRH education, starting at a younger age. SRH education has proven to be effective in increasing SRH knowledge, utilization of MC and decreasing adolescent pregnancies [28]. SRH education can help to prevent risky sexual behaviour and may facilitate access to SRH services.

Youth-friendly services are available in Vientiane and Savannakhet; the main barrier in accessing YFHCs was that youth were unaware of their existence. Our findings are consistent with previous research from Laos [29] and other countries [30, 31, 32] which revealed that unmarried youth said they preferred to get SRH services from private clinics and drugstores, and not public health facilities. In part, the inadequate awareness of the YFHCs was a result of poor promotion of those services. To reach all Lao youth, the services should be more widely available and promoted at schools and in communities. A literature review by Kesterton and Mello in 2010 described promising results of formal referral networks between schools and YFHCs when key figures in schools were trained to inform and refer youth to the SRH services

[33]. The authors suggested that referral systems between community based education programs and YFHCs are also effective in reaching out-of-school youth.

## Psychosocial accessibility

Psychosocial accessibility, especially feeling shy or ashamed because of prevailing negative cultural attitudes towards premarital sex, was perceived as the main barrier for Lao youth to access SRH services. Mostly, they feared that their parents would find out about their visit to the clinic, due to lack of privacy in the health facility, and a perceived lack of confidentiality. Similar findings have been reported elsewhere [4, 34, 14]. Especially in cultures where premarital sex is not socially accepted, youth is reluctant to use SRH services out of fear of exposure. Separate days for youth or re-designed waiting areas could improve privacy in SRH services. To improve social acceptance of youth using SRH services, the youth themselves proposed a need for community projects focused on parents' attitudes. A literature review by Denno, Hoopes, and Chandra-Mouli in 2015 revealed that community support activities have achieved positive results [12, 3]. Another review found similar results, and recommended integrating community activities in multi-component strategies, as community support may facilitate youth access to SRH services [33].

## Geographical accessibility

Youth reported a need for more YFHCs, separate from existing health facilities, to guarantee privacy, in contrast to an earlier finding in Vanuatu, where youth did not value separate clinics [34]. Also, Mazur, Brindis, and Decker discovered that youth in Kenya and Zimbabwe did not want separate youth clinics or single-sex clinics, out of fear of being recognised when they access such services [35]. Given the financial and logistical challenges of separate clinics, further research is needed to determine the best location for YFHCs in Lao PDR. The cost-effectiveness the YFHCs in Vientiane and Savannakhet should be compared, as one stands alone and the other is integrated in a district hospital. Lao youth suggested combining new YFHCs with other youth activity centers. However, reviews of reported studies suggested that may not be an effective approach [33, 12, 3], because youth centers are often used by older male youth for recreation, which could discourage younger males and young women seeking SRH information or services.

## Quality of services

Currently, the Lao government aims to improve the SRH of Lao youth by focusing on enhancing the quality of public SRH services [36]. However, our findings suggest that the barrier for the youth is not a perception of low quality, although they do not actually use the services at present. It seems premature to focus on quality when an emphasis could be placed on alternative ways to provide SRH services to youth. Currently when youth identify an SRH problem, they prefer to visit a pharmacy; public SRH services would only be used when youth were not familiar with the YFHCs and when SRH issues could not be solved outside the SRH clinic. These findings are consistent with those of Tangmunkongvorakul et al, who did similar research among adolescents aged 17–20 years in Northern Thailand. They discovered that adolescents use governmental health services only as a last resort. Furthermore, they noted that adolescents preferred to consult friends when they had SRH problems, and used pharmacies to obtain contraceptives and drugs for self-treatment of STIs [37]. As Lao youth does not access the public SRH services, adapting these services to the needs of Lao youth would not appear to be the most effective way to improve utilisation. Other options should be considered. For example, in Bolivia, where youth used pharmacies in a similar way, the government

responded by training pharmacists to provide youth-friendly counselling. Although long-term effects were not assessed, this did result in an increased demand of SRH services and in better SRH information provision to youth [4]. The applicability of this solution in Lao PDR would need to be investigated. Several efforts have begun in Lao PDR to give youth greater access to SHR information through channels such as a website, a Facebook page and mobile apps. This work does need to be better coordinated to ensure that it is effective; none of it has been evaluated. Nonetheless, confidentiality and privacy issues were also mentioned by adolescents as important components of YFS, which should be based on the real needs of youth rather than only on providers perceptions. By incorporating the criteria of YFHS into health services, use of services by young people has increased [38]. Confidentiality assurances were associated with willingness to seek future health care for routine health needs and to disclose sensitive information (including information related to sexual health), increased frequency of reporting having talked with a provider about sex-related topics in the past, and trust in the provider to keep services related to sexual health issues confidential [39].

The research reported here has strengths and limitations. A well-rounded picture has been obtained by asking about the perceptions of both supply and demand sides. Also, the variety of youth interviewed in terms of schooling, place of residence, sexual activity and sexual orientation, was good. A limitation is that respondents from only two provinces were recruited, both of them major urban areas which will affect the representativeness of the results and their applicability to adolescents living in other parts of Lao PDR, for whom the situation may be more limited than for those in and near the cities. We did not recruit adolescents from other villages than those surrounding the youth clinic, although there may have been experienced and non-experienced adolescents in villages further away.

## Conclusion

Youth in two provinces of Lao PDR experienced a variety of barriers that limited their access to SRH information and services. These insights would be useful to policy makers to improve access to SRH services for youth in Lao PDR.

Recommendations to improve access by youth include:

- A multi-component strategy should be developed including: promotion of YFHCs in the community; a formal referral system between the community and the YFHCs; and community support interventions.

- The government and its partners could consider a SRH mobile clinic in cooperation with school health services, to be taken to schools on specific days as an interim measure to developing youth friendly services.

- Existing FP and SRH services are not promoted at all to adolescents and youth, so links between information and services should be strengthened. The youth friendly health services should be integrated into existing reproductive health services at provincial, district and health centre levels.

- Research is needed on the applicability of these methods in Lao PDR.

## Supporting information

**S1 File.**
(DOCX)

## Acknowledgments

The authors highly appreciate the kind assistance of the University of Health Sciences of Lao PDR, and the contributions from everyone in the research team. Special acknowledgment is extended to Prof. Pamela Wright, who contributed a lot on editing and knowledge of scientific writing.

## Author Contributions

**Conceptualization:** Souksamone Thongmixay, Dirk Rombout Essink, Tim de Greeuw, Viengnakhone Vongxay, Vanphanom Sychareun, Jacqueline E. W. Broerse.

**Data curation:** Souksamone Thongmixay, Tim de Greeuw, Viengnakhone Vongxay.

**Formal analysis:** Souksamone Thongmixay, Tim de Greeuw.

**Methodology:** Souksamone Thongmixay, Dirk Rombout Essink, Tim de Greeuw, Viengnakhone Vongxay, Vanphanom Sychareun, Jacqueline E. W. Broerse.

**Project administration:** Souksamone Thongmixay.

**Supervision:** Dirk Rombout Essink, Vanphanom Sychareun, Jacqueline E. W. Broerse.

**Writing – original draft:** Souksamone Thongmixay, Tim de Greeuw.

**Writing – review & editing:** Souksamone Thongmixay.

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
