## [Decision Letter · Decision Letter 0]

23 Jul 2019

PONE-D-19-14849

Perceived Barriers in Accessing Sexual and Reproductive Health Services for Youth in Lao People’s Democratic Republic

PLOS ONE

Dear Mrs Thongmixay,

Thank you for submitting your manuscript to PLOS ONE. After careful consideration, we feel that it has merit but does not fully meet PLOS ONE’s publication criteria as it currently stands. Therefore, we invite you to submit a revised version of the manuscript that addresses the points raised during the review process.

We would appreciate receiving your revised manuscript by Sep 06 2019 11:59PM. To enhance the reproducibility of your results, we recommend that if applicable you deposit your laboratory protocols in protocols.io, where a protocol can be assigned its own identifier (DOI) such that it can be cited independently in the future. For instructions see: http://journals.plos.org/plosone/s/submission-guidelines#loc-laboratory-protocols

We look forward to receiving your revised manuscript.

Kind regards,

Violet Naanyu

Academic Editor

PLOS ONE

Journal Requirements:

3. Please include your tables as part of your main manuscript and remove the individual files. Please note that supplementary tables (should remain/ be uploaded) as separate "supporting information" files

4. Please provide additional details regarding participant consent.  We note that parental consent for underage participants was not obtained; please confirm whether the IRB approved this form of consent. Moreover, please correct your attached Consetn form, which at present contains the name of the first author.

Reviewers' comments:

Reviewer's Responses to Questions

**Comments to the Author**

1. Is the manuscript technically sound, and do the data support the conclusions?

Reviewer #1: Partly

Reviewer #2: Yes

2. Has the statistical analysis been performed appropriately and rigorously? 

Reviewer #1: Yes

Reviewer #2: Yes

3. Have the authors made all data underlying the findings in their manuscript fully available?

Reviewer #1: No

Reviewer #2: Yes

4. Is the manuscript presented in an intelligible fashion and written in standard English?

Reviewer #1: No

Reviewer #2: Yes

5. Review Comments to the Author

Reviewer #1: General:

• There are a number of grammatical errors across the entire paper. This needs to be addressed

Introduction:

• It would have been valuable for the investigators to give the readers a global and an Asia perceptive on assess to sexual and reproductive health care services before focusing on the situation in Lao PDR. Systematic reviews could be useful for this… This would help the readers gain better insight on the topic of focus.

Methods:

• The methods section does not flow well. There is need to better organize the methods section to enhance flow of content. The investigators could consider the following flow: Study design, Study site, Study population, Sample size and sampling technique, Data collection tool, Data collection procedure (including ethical consideration, recruitment, data collection processes) and Data analysis.

• While doing this, it is important to provide relevant content for each section. As it is, the study design starts with the information that would be best placed in the study population section.

• In the study tools, the investigators mention that the interview guides were based on concepts of the theoretical model yet there is not reference to which theoretical model being applied… Which theoretical model is this? What are the constructs? Are you adapting or adopting all the constructs of the model??

• In the abstract there is mention of semi-structured interview guides being used in the study. However in the study tools section of the manuscript there is no mention of structured questions…. All quest

• In the location section, it would have been useful to provide statistics on the 2 study sites (Vientiane and Savannakhet)… What is the population of adolescents in the 2 study sites? What are the adolescents sexual debut/pregnancy rates in the 2 regions? Given the focus on the study this information would help the readers understand the 2 study sites better and provide an even stronger justification for including the 2 sites.

• In the sample and sampling technique the investigators do not state how they came up with the target numbers of 22 young people and 7 health providers. The aim of qualitative studies is to achieve saturation. Was this achieved? It was also interesting that the investigators did not stratify the sample based on having accessed SRH services. This would be an important strata for this study prior to recruitment… As it is there were only 4 out of 18 young participants who had no SRH experience….Hence the main question is whether the 4 young persons were adequate to achieve saturation for this category of participants….

• It would have also be important to provide information about the interviewers…. What qualifications did they have? Were they trained prior to data collection?

• In the ethical approval the investigator state that they had 3 participants below 18 years and that these participants felt that they were able to give consent…. What does this mean? What are the laws in Lao regarding engaging minors in studies? Were this emancipated minors?

Results

• Table 1 could be presented better. A simple descriptive table with the median age of participants, level of education, sexual encounter and accessed SRH services would have sufficed.

• There is need to organize the themes within in the results section better.

• For example the first theme should be uptake of SRH services. Under this theme subthemes should be, perception about sexual debut, access to SRH, conceptive use, and SRH information channels. There second theme should be barriers to SRH services with focus on barriers only

• Under the sub-theme SRH services the investigators highlight health facility barriers i.e lack or privacy, gender of the provider. Yet the investigators did not consider these are barriers to SRH services. I would therefore recommend that these be moved to barriers

• It is surprising the even under the theme barriers to access of SRH services, there is mention of positive aspects. For example, the investigators state that the young people did not report negative provider attitudes, cost related issues, geographical accessibility, and aspects of quality of services…. Given that the theme focuses on barriers, why are the investigators mentioning these positive aspects?? This is a bit confusing.

• Also confusing is that under geographical accessibility there is mention that it was not perceived as a barrier. Then later the investigators state that it was a barrier among the rural population. Which is which? It was also surprising that under geographical accessibility they mention insufficient services. Insufficient services is not the same as geographical accessibility. Insufficient services means that the facilities are available but they are not providing adequate services or the services are not tailored to the needs of the target population.

• I wish the investigators applied the socioecological model to explore the barriers to adolescent accessing SRH services. This would have provided a more comprehensive perceptive of barriers from the intrapersonal, interpersonal, community, facility, and policy levels

Reviewer #2: PONE-D-19-14849

The article is well written and addresses an important public health issue. The authors assessed perceptions of barriers to youth friendly services. The article should be accepted with the following minor revisions:

Introduction

- The acronym YFHC is spelt out in two ways (youth-friendly health centres/ only two Youth Friendly Health Clinic). The authors should be consistent.

- The objective of the study is stated as “to explore their perceptions regarding accessibility of SRH services for youth” while the title focussed on “barriers”. The word barriers is more generally and could be a better term to use here and the rest of the article.

Methods

The first line of the section on study tools cites table 1 as the study tool yet table 1 in the document actually is a description of study participants. This error should be corrected.

“A semi-structured interview guide was developed for each participant group (Table 1).”

The terms “cognitive accessibility”, “psychosocial accessibility”, “geographical accessibility”, ‘affordability” and “quality of services” should be defined.

The vignettes used in the study could be made available for readers.

Ethical approval – This study recruited youth who were “under the age of majority”. It is not clear if this inclusion of these under age participants without parental consent was approved by the ethical review committee.

Results

The authors report that “The majority of adolescents did not report fearing negative attitudes of health providers in public SRH services. Even though health providers are from an older generation, adolescents assumed that they were professionals who would not treat adolescents in a negative way” yet later report that the adolescents were concerned about lack of confidentiality. It is possible to assume that this perceived lack of confidentiality could be related to the age of the health care providers. The health care providers may find it easy to report them to their parents since they were of the same generation and shared views on adolescent sexuality. The authors later acknowledge that this is the reason why 4 adolescents sought SRH services outside their own district.

Quality of services are ruled out as being of concern to the youths. Privacy and confidentiality are important dimensions of quality. This could be clearer if the authors defined what the elements of quality youth friendly services are.

Discussion

“Youth-friendly services are available in Vientiane and Savannakhet; the main barrier in accessing YFHCs was being unaware of their existence.” - this statement needs to be qualified – there are only two YFHCs. The accuracy of this statement can only be assessed if information on the geographical size and population of the two services is presented.

The authors rule out the role of quality of youth friendly services as being a barrier yet acknowledged in the results the concerns of privacy and confidentiality which could actually be elements of quality. The authors should reconsider this statement.

The role of lack of privacy and confidentiality has not been adequately discussed. This could be a key barrier to YFHC services.

Recommendations

There are recommendations about provision of youth friendly services in schools which is not adequately supported by the data present

6. PLOS authors have the option to publish the peer review history of their article (what does this mean?). If published, this will include your full peer review and any attached files.

Reviewer #1: No

Reviewer #2: Yes: Dickens Onyango

---

## [Author Response · Author response to Decision Letter 0]

5 Sep 2019

Dear Reviewer and editor,

Thank you for your question and comment, I have added and made changed based on reviewer suggestion. I have uploaded my file in attach files

A rebuttal letter that responds to each point raised by the academic editor and reviewer(s). I uploaded as separate file and labeled 'Response to Reviewers'.

A marked-up copy of your manuscript that highlights changes made to the original version. I uploaded as separate file and labeled 'Revised Manuscript with Track Changes'.

An unmarked version of your revised paper without tracked changes. I uploaded as separate file and labeled 'Manuscript'.

With best regards,

Souksamone

---

## [Decision Letter · Decision Letter 1]

23 Sep 2019

Perceived Barriers in Accessing Sexual and Reproductive Health Services for Youth in Lao People’s Democratic Republic

PONE-D-19-14849R1

Dear Dr. Souksamone Thongmixay,

We are pleased to inform you that your manuscript has been judged scientifically suitable for publication and will be formally accepted for publication once it complies with all outstanding technical requirements.

With kind regards,

Violet Naanyu

Academic Editor

PLOS ONE

Reviewers' comments:

Reviewer's Responses to Questions

**Comments to the Author**

1. If the authors have adequately addressed your comments raised in a previous round of review and you feel that this manuscript is now acceptable for publication, you may indicate that here to bypass the “Comments to the Author” section, enter your conflict of interest statement in the “Confidential to Editor” section, and submit your "Accept" recommendation.

Reviewer #1: All comments have been addressed

Reviewer #2: All comments have been addressed

2. Is the manuscript technically sound, and do the data support the conclusions?

Reviewer #1: Yes

Reviewer #2: Yes

3. Has the statistical analysis been performed appropriately and rigorously? 

Reviewer #1: N/A

Reviewer #2: Yes

4. Have the authors made all data underlying the findings in their manuscript fully available?

Reviewer #1: Yes

Reviewer #2: Yes

5. Is the manuscript presented in an intelligible fashion and written in standard English?

Reviewer #1: Yes

Reviewer #2: Yes

6. Review Comments to the Author

Reviewer #1: I feel that the reviewers have addressed most of my comments. I have no additional comments on this manuscript.

Reviewer #2: The article is well written, authors have revised the manuscript and addressed concerns raised by both reviewers.

7. PLOS authors have the option to publish the peer review history of their article (what does this mean?). If published, this will include your full peer review and any attached files.

Reviewer #1: No

Reviewer #2: Yes: Dickens Onyango

---

## [Editor Report · Acceptance letter]

14 Oct 2019

PONE-D-19-14849R1 

Perceived Barriers in Accessing Sexual and Reproductive Health Services for Youth in Lao People’s Democratic Republic 

Dear Dr. Thongmixay:

I am pleased to inform you that your manuscript has been deemed suitable for publication in PLOS ONE. Congratulations! Your manuscript is now with our production department. 

With kind regards,

on behalf of

Prof. Violet Naanyu 

Academic Editor

PLOS ONE